# The Impact of DNMT3A Status on NPM1 MRD Predictive Value and Survival in Elderly AML Patients Treated Intensively

**DOI:** 10.3390/cancers13092156

**Published:** 2021-04-29

**Authors:** Maël Heiblig, Nicolas Duployez, Alice Marceau, Delphine Lebon, Laure Goursaud, Isabelle Plantier, Laure Stalnikiewich, Nathalie Cambier, Marie Balsat, Gaëlle Fossard, Hélène Labussière-Wallet, Fiorenza Barraco, Sophie Ducastelle-Lepretre, Pierre Sujobert, Sarah Huet, Sandrine Hayette, Hervé Ghesquières, Xavier Thomas, Claude Preudhomme

**Affiliations:** 1Hospices Civils de Lyon, Hematology Department, Lyon-Sud Hospital, 69310 Pierre Bénite, France; marie.balsat01@chu-lyon.fr (M.B.); gaelle.fossard@chu-lyon.fr (G.F.); helene.labussiere-wallet@chu-lyon.fr (H.L.-W.); fiorenza.barraco@chu-lyon.fr (F.B.); sophie.ducastelle-lepretre@chu-lyon.fr (S.D.-L.); pierre.sujobert@chu-lyon.fr (P.S.); sarah.huet@chu-lyon.fr (S.H.); sandrine.hayette@chu-lyon.fr (S.H.); herve.ghesquieres@chu-lyon.fr (H.G.); xavier.thomas@chu-lyon.fr (X.T.); 2Department of Hematology, Lyon-Sud Hospital, Bat. 1G, 165 chemin du Grand Revoyet, CEDEX, 69310 Pierre Bénite, France; 3UMR9020–UMR-S 1277—Canther—Cancer Heterogeneity, Plasticity and Resistance to Therapies, Institut de Recherche contre le Cancer de Lille, University of Lille, CNRS, Inserm, CHU Lille, 59000 Lille, France; nicolas.duployez@chru-lille.fr (N.D.); alice.renaut@chru-lille.fr (A.M.); claude.predhomme@chru-lille.fr (C.P.); 4Hematology Department, CHU Amiens, 80000 Amiens, France; lebon.delphine@chu-amiens.fr; 5Center for Infection and Immunity of Lille, Hematology Department, 59019 Lille, France; laure.GOURSAUD@CHRU-LILLE.FR; 6Hematology Department, Roubaix Hospital, 59512 Roubaix, France; isabelle.plantier@ch-roubaix.fr; 7Hematology Department, Lens Hospital, 62300 Lens, France; laure.stalnikiewich@ch-lens.fr; 8Hematology Department, Valenciennes Hospital, 59300 Valencienne, France; cambier-n@ch-valenciennes.fr

**Keywords:** acute myeloid leukemia, elderly, prognosis, *NPM1*, *DNMT3A*

## Abstract

**Simple Summary:**

*DNMT3A* mutation has been associated with adverse outcomes. In this study, we aimed to investigate the impact of *DNMT3A* status on *NPM1* MRD predictive value for survival in a retrospective cohort of acute myeloid leukemia (AML) patients aged over 60 years old treated intensively. A total of 138 patients treated for *NPM1*-mutated AML in two French institutions were analyzed retrospectively. A 4log reduction of *NPM1* MRD was associated with a better outcome. *DNMT3A* negative patients who achieved a 4log reduction had a superior outcome to those who did not. However, postinduction *NPM1* MRD1 reduction was not predictive of OS and LFS in *DNMT3A*mut patients. These results confirm that post-induction *NPM1* MRD1 is a reliable tool to assess disease outcome in elderly AML patients. However, the presence of *DNMT3A* also identify a subgroup of patients at high risk of relapse.

**Abstract:**

Minimal residual disease (MRD) is now a powerful surrogate marker to assess the response to chemotherapy in acute myeloid leukemia (AML). *DNMT3A* mutation has been associated with adverse outcomes. In this study, we aimed to investigate the impact of *DNMT3A* status on NPM1 MRD predictive value for survival in a retrospective cohort of AML patients aged over 60 years old treated intensively. A total of 138 patients treated for *NPM1*-mutated AML in two French institutions were analyzed retrospectively. *DNMT3A* status did not influence the probability of having a ≥ 4log MRD1 reduction after induction. Only 20.4% of *FLT3-ITD* patients reached ≥ 4log MRD1 reduction compared to 47.5% in *FLT3*wt cases. A 4log reduction of *NPM1* MRD was associated with a better outcome, even in *FLT3-ITD* mutated patients, independent of the allelic ratio. *DNMT3A* negative patients who reached a 4log reduction had a superior outcome to those who did not (HR = 0.23; *p* < 0.001). However, postinduction *NPM1* MRD1 reduction was not predictive of OS and LFS in *DNMT3A*mut patients. These results confirm that post-induction *NPM1* MRD1 is a reliable tool to assess disease outcome in elderly AML patients. However, the presence of *DNMT3A* also identifies a subgroup of patients at high risk of relapse.

## 1. Introduction

Although the outcome in younger adults with acute myeloid leukemia (AML) has improved these last decades, the treatment of elderly patients remains challenging [1]. Indeed, recent data suggest that the European LeukemiaNet (ELN) 2017 risk stratification per se was not accurate to stratify properly patients older than 60 years old treated with intensive chemotherapy [2]. AML patients over 60 years carried adverse cytogenetics more frequently than younger adults but also a specific gene-expression and molecular alterations that support a molecular basis for poor outcomes [3]. Moreover, approximately 70% of patients over 60 years carry unfavorable cytogenetic or molecular markers whereas only 30 to 35% of them are effectively transplanted [4,5]. This might be explained by the fact that very few patients over the age of 70 years old reach transplantation due to excess toxicity during induction, comorbidities, or a less complete remission rate. In the absence of hematopoietic stem cell transplantation (HSCT), it is still unclear if these patients may benefit from intensive chemotherapy, even in the ELN 2017 favorable risk group, over alternative strategies, especially in the era of venetoclax-based combination [6]. 

Minimal residual disease is now a powerful surrogate marker to assess the response to chemotherapy. Several techniques are used routinely to assess MRD such as RT-qPCR, flow cytometry, and next generation sequencing (NGS). However, only *NPM1* and CBF MRD by RT-qPCR are considered reliable tools to re-allocate patients into different risk groups. In younger adults, *NPM1* MRD has recently been demonstrated as a favorable predictive marker for event-free survival (EFS) and overall survival (OS) independent of fms-like tyrosine kinase-3 internal tandem duplications (*FLT3-ITD*) status. Balsat et al. [7] prospectively showed in a cohort of 229 patients aged below 60 that a 4-log reduction of *NPM1* MRD after induction conferred a better survival, independent of *FLT3-ITD* status. These results were in line with those previously published by Krönke et al. [8]. However, there are very few data regarding the predictive value of NPM1 MRD in elderly AML patients (aged over 60 years old). In elderly patients, it is also rather unclear whether good molecular responders or at least those with undetectable MRD by flow-cytometry may benefit from allogenic HSCT, especially in the intermediate subgroup [9]. Numerous studies have suggested the negative impact of *DNMT3A* mutation in NPM1-mutated AML patients, especially in those with concurrent *FLT3-ITD* mutation [2,10,11]. 

In this study, we aimed to investigate the impact of *DNMT3A* status on *NPM1* MRD predictive value for survival in a retrospective cohort of AML patients aged over 60 years old treated intensively.

## 2. Patients and Methods

### 2.1. Patients

A total of 138 patients (aged 60 years or more) treated in Lyon and Hauts de France centers (CHU Lille, CHU Amiens, Hôpital St Vincent, CH Valenciennes, CH Arras, CH Boulogne-sur-mer, CH Roubaix, CH Lens, CH Dunkerque), France, with newly diagnosed *NPM1*-mutated AML (de novo or secondary) for which post induction bone marrow (BM) minimal residual disease (MRD) were available, were included in this study. At diagnosis, blood and BM samples were examined for cytogenetic abnormalities and molecular markers (*NPM1, FLT3-ITD, DNMT3A*) according to local procedures. Patients were assigned to risk groups according to the 2017 ELN classification, regarding the *FLT3-ITD* ratio [12]. Karyotype was not considered in the risk stratification. As the *TP53*, *ASXL1*, and *RUNX1* mutational status were not available, these variables were not included in the risk stratification.

### 2.2. Treatments

All patients were treated by an anthracycline- and cytarabine-based induction chemotherapy regimen. One hundred and fifteen of them were treated in or according to the observational Acute Leukemia French Association (ALFA)-1200 study, with or without midostaurin (depending on the presence of *FLT3-ITD* and drug availability). The 23 other patients were included in other protocols (eight in ALFA-0701, 10 in MyloFrance 4 (NCT02473146), three in BIG1 protocol (NCT02416388)], one in ALFA-0702, one in BRIGHT protocol (NCT03416179) [13,14]. Patients achieving composite complete remission (CRc) after one or two courses of induction were given consolidation chemotherapy according to the protocol in which they were included. Nineteen patients underwent HSCT in their first CRc after reduced intensity conditioning (RIC).

### 2.3. Clinical and Molecular Markers

Cytogenetic analysis was performed according to the International System for Human Cytogenetic Nomenclature guidelines [15]. *NPM1* and *FLT3-ITD* mutations were detected and quantified as previously described [16,17]. MRD for *NPM1* was assessed after at least the first cycle of induction (MRD1). Quantification of the different *NPM1* mutation types was performed by allele-specific oligonucleotide real-time quantitative polymerase chain reaction (ASO-RQ-PCR) using a common forward primer, a mutation-specific reverse primer and a common hydrolysis probe, as previously described [18,19,20,21,22]. The *NPM1*m copy number value was then normalized on the number of ABL1 transcripts used as an endogenous reference gene [23,24,25]. All patients with *NPM1*, *FLT3-ITD* status, and at least post-induction *NPM1* MRD on BM were included in this retrospective study. Patients with atypical *NPM1* transcripts were also include into the descriptive analysis. Only patients with monitorable transcripts were included in the survival analysis.

### 2.4. Outcome Parameters

CRc (CR + Cri + CRp) status was defined on BM aspirates with less than 5% of blasts recovery and classical hematological recovery characteristics [12]. Overall survival (OS) was calculated from treatment assignment to death from any cause. Leukemia-free survival (LFS) was determined for responders from CRc until disease relapse or death of any cause. Living patients were censored for OS at the last follow-up date, and patients in CR were censored for LFS at the last disease assessment.

### 2.5. Statistical Analyses

Comparative descriptive statistics were used to characterize patients and their disease in their entirety. Continuous variables were reported as the median ± standard deviation (SD) followed by a *t* test if the distribution was normal in both groups; if not, the median, range (min-max), and inter-quartile range (IQR) were assessed with a Mann–Whitney test to compare groups. The discrete and qualitative variables were reported as count and percentage. The probabilities of LFS and OS were estimated using the Kaplan–Meier method, and the log-rank rest evaluated the differences between survival distributions. Univariate and multivariate analyses including the baseline demographic, and clinical and molecular features were studied thanks to Cox regressions. The statistical results were two-sided with a *p*-value < 0.05 considered statistically significant (XLSTAT©).

## 3. Results

### 3.1. Initial Patient Characteristics

The median age of the entire cohort was 66.1 years old (range: 60 to 78.2). Table 1 shows the distribution of patient’s characteristics by *DNMT3A* status. Of the 138 *NPM1**mut* patients, *DNMT3A* status was available in 98 of them. Overall, 10 out of 138 patients were considered to have a secondary AML (previous history of myelodysplastic syndrome or therapy-related AML). The most frequent *DNMT3A* alteration was R882 missense mutation (51.9%). *DNMT3A* mutations were usually part of the main clone with a median variant allelic frequency of 42% (Table 1). *FLT3-ITD* was evidenced in 52 of 138 patients (37.6%) with a median *FLT3-ITD* allele ratio (AR) of 0.53 (range: 0.05 to 3). *FLT3-ITD* mutations co-occurred more frequently in *DNMT3A**mut* patients (48.1% vs. 21.4%, *p* < 0.001). In this cohort, 40 patients were classified in the unfavorable ELN subgroup because of the presence of *FLT3-ITD*. Higher median *NPM1* baseline levels were detected in cases with *DNMT3A*mut and *FLT3-ITD* compared to double negative cases (1384% vs. 1685%; *p* = 0.045) (Appendix A). No significant correlation was found between the *DNMT3A* mutational status and age, *FLT3-ITD* allelic ratio (AR), or karyotype.

### 3.2. General Outcome According to DNMT3A and FLT3-ITD Status

With a median follow-up of 20 months (0.07 to 128.4), the overall CRc rate was 89.9% with no influence of *DNMT3A* or *FLT3* mutational status on the probability of CR. MRD1 response was available in 126/138 patients at the end of induction. The median OS of the entire cohort was 30.6 months with an OS rate of 69.1% at 1 year and 47.5% at 3 years. LFS was 59.1% at 1 year and 33.5% at 3 years (median PFS: 15.4 months) (Appendix A). The presence of *FLT3-ITD* mutation was associated with a worse OS (median OS: 10.9 months) compared to *NPM1*_mut_ FLT3_wt_ patients (median OS: 46.3 months, *p* < 0.001) (Figure 1A). There was no significant difference in OS and LFS between *NPM1*_mut_ FLT3_wt_ and *NPM1*_mut_ *FLT3-ITD*_low_. Even if there was no difference between *NPM1*_mut_ *FLT3-ITD*_high_ (median OS = 10.6 months) and *NPM1*_mut_ *FLT3-ITD*_low_ (median OS = 22.6 months), patients with high *FLT3-ITD* AR have a worse outcome compared to wild type ones (Figure 1B, Appendix A). Regarding the prognostic impact of *DNMT3A* mutation, the presence of the mutation was associated to a worse outcome in terms of OS (*DNMT3A*_mut_ vs. *DNMT3A*_wt_ median OS: 18.5 months vs. not reached (NR), *p* = 0.002) and LFS (*DNMT3A*_mut_ vs. *DNMT3A*_wt_ median LFS: 9.3 months vs. 54.7 months, *p* < 0.001) (Figure 1C, Appendix A). Among the 52 patients having *DNMT3A* and *FLT3-ITD* status available, the presence of *DNMT3A*_mut_ and *FLT3-ITD* significantly worsen the OS and LFS compared to isolated *NPM1*-mutated patients. In *NPM1*_mut_ *DNMT3A*_mut_ cases, *FLT3-ITD* impaired significantly OS and LFS (Figure 1D, Appendix A).

### 3.3. Prognostic Impact of Postinduction NPM1 BM-MRD Log Reduction

Postinduction *NPM1* MRD1 on BM was available in 127 patients (92%). In this elderly cohort of *NPM1*_mut_ patients, a 4log reduction of *NPM1* MRD was associated with a better outcome in terms of OS (median OS: NR vs. 13.4 months, HR = 0.35, *p* < 0.01) and LFS (Figure 2A, Appendix A). Similarly, a 5log reduction identifies a subgroup of patients with a very favorable outcome when compared to a 4log reduction (Figure 2B). Overall, *DNMT3A* status did not influence the probability of having a ≥4log MRD1 reduction after induction. However, only nine out of 44 (20.4%) *FLT3-ITD* patients reached ≥4log MRD1 reduction whereas 38 out of 80 FLT3_wt_ (47.5%) were good molecular responders (*p* < 0.001). A high *FLT3-ITD* allelic ratio or *DNMT3A* co-occurrence have no significant impact on the probability of reaching a ≥ 4log MRD1 reduction. *FLT3-ITD*-mutated patients who achieved a 4log reduction had a superior outcome to those who did not (HR = 0.34; 95% CI, 0.16 to 0.70; *p* < 0.001). Similarly, *NPM1*_mut_ FLT3_wt_ patients with a 4log reduction in *NPM1* BM-MRD had a longer OS (three-year OS, 68.1%; 95% CI, 48.8 to 82.9) than those without good molecular response (three-year OS, 46.5%; 95% CI, 30.2 to 61.7) (Figure 2C). *DNMT3A* negative patients who achieved a 4log reduction had a superior outcome to those who did not reached at least a 4log reduction (HR = 0.23; 95% CI, 0.07 to 0.72; *p* < 0.001). However, postinduction *NPM1* MRD1 reduction was not predictive of OS in *DNMT3A* positive patients (Figure 2D). *DNMT3A*_mut_ patients had a very poor LFS which was even worse in poor *NPM1* MRD1 responders compared to those who reached at least a 4log reduction (Appendix A).

### 3.4. Multivariate Analysis and Prognosis Model

In the univariate analysis for OS, the variables associated with a poorer survival were the presence of *FLT3-ITD* mutation, *DNMT3A* mutation, absence of MRD1 4log reduction and unfavorable ELN risk group. Age, HSCT in CR1 and karyotype were not of prognostic value (Appendix A). In multivariate analysis, only the *DNMT3A* mutational status and a 4-log reduction in *NPM1* BM-MRD were significantly associated with survival (Table 2).

Based on these results, we identified among *NPM1* positive patients three groups with distinct prognosis, based on *FLT3-ITD*, *DNMT3A* status, and the *NPM1* BM-MRD post-induction response (Figure 3A). Within the ELN 2017 favorable risk group (*n* = 98), the NPM1 scoring system (NPM1 SS) reallocated 45.9 and 25.5% of patients into intermediate and unfavorable risk groups, respectively (Figure 3B). The median OS of the favorable, intermediate, and unfavorable NPM1 SS risk groups were NR, 30.6, and 13.2 months, respectively (Figure 3C). The median LFS of the favorable, intermediate, and unfavorable NPM1 SS risk group were NR, 17.2, and 7.7 months, respectively (Figure 3D, Appendix A). We then used ROC curve comparison to assess if NPM1 SS could be more accurate in OS prediction than ELN classification. When compared to ELN (AUC = 0.695), the NPM1 scoring system was more accurate for OS prediction in patients within the intermediate (AUC = 0.833) and unfavorable (AUC = 0.863) NPM1 SS risk group. However, there was no significant difference in AUC between the NPM1 SS favorable risk group and ELN favorable risk group (Figure 3E).

## 4. Discussion

The management of AML in patients older than 60 years remains a major challenge as it is still rather unclear which patients will benefit or not from intensive chemotherapy compared to low-intensity regimens according to clinical, molecular, or cytogenetics markers, especially in the absence of HSCT [9]. Several factors may be involved in the poor prognosis of older AML patients. Silva et al. reported that elderly AML is a specific and distinct entity that harbors genetic and epigenetic patterns not shared by younger adults [19]. Beside a specific methylation signature, these AMLs are enriched in genetic alterations in spliceosome machinery, epigenetic regulators, and in DNA repair factors known to be associated with global treatment resistance. Aging has also been related to an increased frequency of unfavorable karyotype incidence, probably due to the increasing prevalence of secondary AML [20]. Nevertheless, after 60 years old, Nagel et al. showed that there were no major modifications among the ELN risk group distribution under and beyond 70 years old, which was the same regarding the *NPM1* and *FLT3-ITD* mutations distribution, suggesting that other cofounding factors such as secondary mutations may impact on survival [3]. Indeed, ELN 2017 classification shows some limitation into stratifying patients, especially those within favorable or intermediate subgroups that are not referred systematically to HSCT. Recently, Gardin et al. [13] showed that integrating secondary AML-like gene mutations (*ASXL1, SRSF2, STAG2, BCOR, U2AF1, EZH2, SF3B1,* and *ZRSR2*) identifies a specific subset of high-risk patients and may thus improve the definition of high-risk older patients with AML [13].

*FLT3-ITD* mutations and its high allelic ratio is well known to be associated with a high risk of relapse and dismal outcomes in younger adults, whereas its impact seems to be reduced or absent in elderly patients [23,24,26]. In our study, patients with low *FLT3-ITD* AR have a similar outcome than those without *FLT3* mutation, suggesting that other confounding factors might significantly impact survival. We also showed that the sole presence of *FLT3-ITD* was not an independent prognostic factor among *NPM1*-mutated patients when integrating other molecular markers such as *DNMT3A* and MRD monitoring. Regarding *DNMT3A*, its mutation has been associated with adverse outcomes among patients with an intermediate-risk cytogenetic profile or *FLT3* mutations, regardless of age [11,27,28]. In AML, the exact pathogenic mechanism by which *DNMT3A* mutations act on and negatively affect the outcome is rather unclear. Despite the apparent biological consequences of *DNMT3A* loss in normal hematopoietic and leukemic stem cells (LSC), the correlation between the differentially methylated genes and gene expression have never been demonstrated, suggesting that other mechanisms may be involved [29,30]. Moreover, significant differences in DNA methylation signatures between AML with *DNMT3A*-R882 and non-R882 mutations have been reported, suggesting that mutations in different *DNMT3A* domains lead to different neomorphic functions [31]. Additionally, Guryanova et al. [10,32] showed recently that *DNMT3A* cooperates with *FLT3-ITD* and *NPM1* to induce AML in vivo, and promotes resistance to anthracycline chemotherapy through impaired nucleosome eviction and chromatin remodeling in response to anthracyclines. As with Papaemmanuil et al. [10], our results suggest also that the co-occurrence of *FLT3-ITD* and *DNMT3A* mutations might be detrimental regarding the outcome.

*NPM1* gene mutations are important as molecular markers for the diagnosis, prognosis, and monitoring of MRD in AML patients. Several groups have reported the prognostic impact of the *NPM1* mutation MRD response, especially when assessed early [7,33,34,35,36]. Ivey et al. prospectively showed that *NPM1* MRD performed after the first cycle of consolidation was highly predictive of relapse-free survival (RFS) and OS, independent of the *FLT3-ITD* and *DNMT3A* mutational status [36]. These results were recently confirmed with postinduction MRD, which showed that patients harboring at least a 4-log reduction did not benefit from HSCT, independent of the *FLT3-ITD* status [7]. This is, to our knowledge, the largest study assessing the impact of *NPM1* BM-MRD1 on survival in elderly AML patients treated with intensive chemotherapy. Although multiple studies evaluating *NPM1*_mut_ MRD also included some patients aged > 60 years, none of them specifically focused on the application and clinical significance of MRD monitoring in elderly patients with *NPM1*_mut_ AML [36,37,38]. As in younger adults, we showed that a 4log reduction of *NPM1* BM-MRD1 after induction was of strong prognostic significance, especially in patients with *FLT3-ITD* mutation. More stringent MRD responses such as 5log reduction may be also more accurate to identify patients with a more favorable outcome. Our results suggest that the presence of *DNMT3A* may abrogate the predictive value of *NPM1* BM-MRD1 on survival, at least in the elderly. However, these results should be validated prospectively, especially in younger AML patients.

More recently, Herold et al. [2] showed that ELN 2017 classification could be refined within each risk group by integrating the *DNMT3A* mutational status that systematically identified a subgroup of patients with significantly inferior OS compared with *DNMT3A* wild-type patients. Interestingly, the incidence of *DNMT3A* mutation seems to increase after the age of 40 years old, suggesting that this mutation is critical in leukemogenesis as it is involved in clonal hematopoiesis [39]. By integrating *DNMT3A, NMP1* MRD and *FLT3-ITD* status, we identified among *NPM1*-mutated elderly patients three subgroups with different outcomes. NPM1 SS identifies among ELN a favorable subgroup—elderly patients with dismal prognoses that might benefit from HSCT in their first CR. However, the number of transplanted patients were too small to validate this hypothesis. We should also be cautious about the interpretation of results in this elderly patient population. Although the sample size is fairly large in this specific *NPM1*-mutated cohort of patients, we still cannot be completely confident about the impact of other confounding factors. Limitations in our study mainly concern its retrospective profile, MRD assessment on bone marrow over peripheral blood, and the absence of other molecular parameters such as *TP53* or secondary mutations in the prognostic model.

In conclusion, our work confirms that post-induction *NPM1* MRD1 is a reliable tool to assess the disease outcome in elderly AML patients. Even if these results have some limitations, the presence of *DNMT3A* seems to abrogate the predictive value of *NPM1* good molecular responses, but deserves to be addressed and confirmed in larger studies. Future efforts should focus on identifying older patients who clearly derive a survival benefit from allogeneic transplantation in their first remission. Nevertheless, new therapeutic agents may drastically change the prognosis of AML in elderly patients, especially FLT3 inhibitors (i.e., gilteritinib) and venetoclax [6,40].

## Figures and Tables

**Figure 1 cancers-13-02156-f001:**
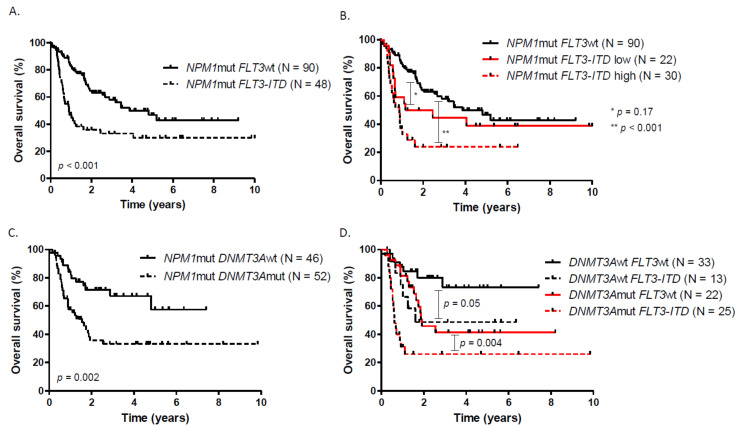
Overall survival according to (**A**) *FLT3-ITD* status, (**B**) *FLT3-ITD* allelic ratio, (**C**) *DNMT3A* status, (**D**) *DNMT3A* and *FLT3-ITD* status.

**Figure 2 cancers-13-02156-f002:**
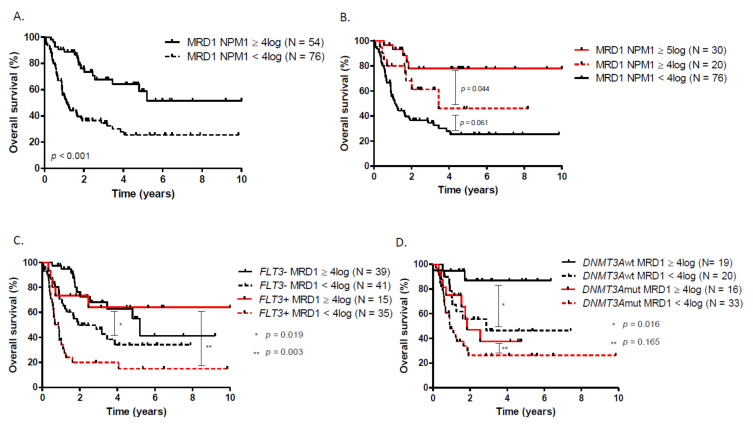
Overall survival according to (**A**) *NPM1* MRD1, (**B**) *NPM1* MRD1 with a 5log reduction cut-off, (**C**) *NPM1* MRD1 in patients with or without *FLT3-ITD*, (**D**) *NPM1* MRD1 in patients with or without *DNMT3A*.

**Figure 3 cancers-13-02156-f003:**
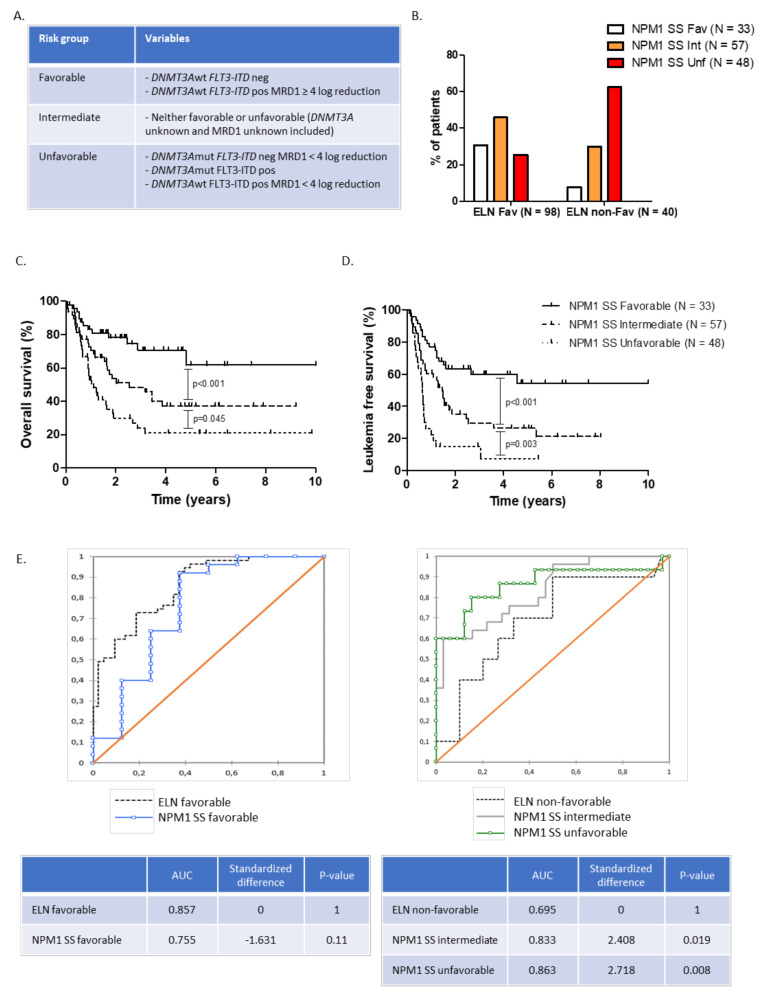
(**A**) NPM1 scoring system (SS) risk groups, (**B**) ELN 2017 risk groups reallocation according to NPM1 SS, (**C**) overall survival, and (**D**) leukemia-free survival in NPM1 SS favorable, intermediate, and unfavorable risk groups. (**E**) ROC curve analysis assessing overall survival prediction according to ELN and NPM1 SS.

**Table 1 cancers-13-02156-t001:** Patient characteristics.

Characteristics	All Cohort (N = 138)	*DNMT3A*mut (N = 52)	*DNMT3A*wt (N = 42)
Age, years, median (range)	66.1 (60–78.2)	65.9 (60.1–76.1)	66.3 (60–76.3)
60–64 yo	-	27/46 (58.7%)	19/46 (41.3%)
65–69 yo	-	14/29 (48.2%)	15/29 (51.8%)
70+ yo	-	11/19 (57.9%)	8/19 (42.1%)
Secondary AML, n(%)	10/138 (7.3%)	-	-
**Karyotype**			
Normal, n(%)	113/138 (81.8%)	44/52 (84.6%)	34/42 (81.0%)
Abnormal intermediate, n(%)	15/138 (10.9%)	5/52 (9.6%)	4/42 (9.5%)
Unfavorable, n(%)	4/138 (2.9%)	3/52 (5.8%)	1/42 (2.4%)
Missing, n(%)	6/138 (4.4%)	0/52	3/42 (7.1%)
**NPM1 type of mutation**			
Type A, n(%)	105/138 (76.1%)	-	-
Type B, n(%)	11/138 (8.0%)		
Type D, n(%)	10/138 (7.3%)		
Others, n(%)	12/138 (8.6%)		
***FLT3*-ITD status**			
Positive, n(%)	52/138 (37.6%)	25/52 (48.1%)	9/42 (21.4%)
Ratio, median (range)	0.53 (0.05–3)	0.59 (0.07–4.3)	0.56 (0.14–0.7)
***DNMT3A* type of mutation**			
Positive, n(%)	52/94 (55.3%)	-	-
R882, n(%)	27/52 (51.9%)		
Missing, n(%)	40/138 (28.9%)		
VAF, median, % (range)	42% (1.8–50)		
***NPM1 MRD1 response (evaluable in 126/138 patients)***			
≥4log reduction, n(%)	50/126 (38.9%)	16/49 (32.6%)	15/35 (42.8%)
<4 log reduction, n(%)	77/126 (61.1%)	33/49 (67.4%)	20/35 (57.2%)
**HSCT in CR1, n(%)**	19/138 (13.7%)	6/52 (11.5%)	11/42 (26.2%)
**Median follow-up, months (range)**	20.0 (0.07–128.4)	-	-

CR = complete response, HSCT = hematopoietic stem cell transplantation, MRD = minimal residual disease, VAF = variant allelic frequency.

**Table 2 cancers-13-02156-t002:** Multivariate analysis.

Variables	Overall Survival HR (IC 95%)	*p*-Value	Leukemia Free Survival HR (IC 95%)	*p*-Value
ELN classification (Favorable vs. other)	1.75 (0.97–3.15)	0.062	1.34 (0.77–2.33)	0.30
*FLT3*-ITD (wt vs. mut)	1.48 (0.81–2.58)	0.21	1.56 (0.92–2.66)	0.097
*DNMT3A* (wt vs. mut)	2.08 (1.06–4.1)	0.034	2.41 (1.33–4.36)	0.004
*DNMT3A* (wt vs. unknown)	2.07 (1.04–4.15)	0.038	2.02 (1.09–3.73)	0.024
MRD1 ≥ 4log reduction (yes vs. no)	2.72 (1.54–4.80)	0.001	2.56 (1.56–4.17)	<0.001

## Data Availability

Not applicable.

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
