# Peer review of "The Impact of DNMT3A Status on NPM1 MRD Predictive Value and Survival in Elderly AML Patients Treated Intensively"

_cancers, 2021, doi:10.3390/cancers13092156_

Round 1

Reviewer 1 Report

The authors have presented their work in a very lucid manner. Their work has identified the role of DNMT3a status on predictive value of the NPM1 MRD in AML patients. In doing so they have made use of the overall survival and leukemia free survival values as a proxy to test the effective predictive value of NPM1 MRD. The work will be therefore prove to be an essential factor in better definition of risk groups and planning of patient treatment regimen, especially for the DNMT3a mut patients. 

The figure 2 was also published by most of the current authors in November 2020 (https://ashpublications.org/blood/article/136/Supplement%201/7/470193/Impact-of-DNMT3a-Status-on-Post-Induction-NPM1-MRD). The results were also summarized in the above mentioned link but is now well elaborated here in this submission.

In my opinion, the manuscript represents the clinical data in a very concise manner and brings forth an interesting finding about DNMT3a in AML relapse predictive potential.

Author Response

Thank you for your comment. 

Reviewer 2 Report

With the manuscript “Impact of DNMT3A status on NPM1 MRD predictive value on survival in elderly AML patients treated intensively” the authors aim at studying if DNMT3A can be used as a marker for AML in patients over 60 years old, where the prognosis is usually worst, and the therapeutic options are reduced. Overall, the data presented suggests that DNMT3A, NPM1 and FLT3 are good markers to stratify patients. However, these finds are not novel and previous work has already shown the importance of DNMT3A, please see Bezerra et al 2020 (Blood), Cappelli et al 2019 (Leukemia), Lachowiez et al 2020 (Blood Adv). The authors need to clarify better their findings.

Comments:

  • Manuscript needs text revision.
  • Several supplementary figures where not available to review; figure supplementary 4 is not mentioned in the text
  • Table 1 should indicate if samples are from de novo or secondary AML

Author Response

With the manuscript “Impact of DNMT3A status on NPM1 MRD predictive value on survival in elderly AML patients treated intensively” the authors aim at studying if DNMT3A can be used as a marker for AML in patients over 60 years old, where the prognosis is usually worst, and the therapeutic options are reduced. Overall, the data presented suggests that DNMT3A, NPM1 and FLT3 are good markers to stratify patients. However, these finds are not novel and previous work has already shown the importance of DNMT3A, please see Bezerra et al 2020 (Blood), Cappelli et al 2019 (Leukemia), Lachowiez et al 2020 (Blood Adv). The authors need to clarify better their findings.

Thank you for your comment. Indeed, the negative impact of DNMT3A and FLT3-ITD co-mutational burden has been previously reported regarding in NPM1 mutated patients. However, most of these retrospective works did not focused on patients aged over 60 years old specifically. Moreover, predictive value of NPM1 MRD have been essentially reported for patients younger than 60 years old. Nowadays, NPM1 MRD1 or 2 is widely used to allocate patients to HSCT or not. By extension, NPM1 MRD predictive value has been used in elderly patients. However, there is currently no published datas in this population subset. This is to our knowledge the first study assessing retrospectively the predictive value of NPM1 MRD in elderly patients. Our study suffers from some limitations such as its retrospective nature and the limited number of patients. Nevertheless, we showed that DNMT3A mutated patients seems to have very poor outcome even in cases of good molecular response during intensive chemotherapy, suggesting that NPM1 MRD might not be a reliable tool in this subset of patients. Discussion on this topic have been improved (line 289-294). Conclusion has also been clarified regarding key message of the manuscript (line 308-314). Suggested references have been added to the manuscript. 

Comments:

  • Manuscript needs text revision.

--> Text revisions have been realized on this final version.

  • Several supplementary figures where not available to review; figure supplementary 4 is not mentioned in the text

--> Supplemental figure 4 reference has been added in the main text. Missing supplemental figures/tables have also been added in the revised version.

  • Table 1 should indicate if samples are from de novo or secondary AML

-->The number of patients with documented history of sAML have been implemented in the main text (line 149) and in table 1.

General comments

Introduction have been improved (line 34-48) and a simple summary (line 24-33) added to the manuscript.

Reviewer 3 Report

In their article “Impact of DNMT3A status on NPM1 MRD predictive value on survival in elderly AML patients treated intensively” Heiblig et al. present a retrospective cohort study on the prognostic impact of NPM1 MRD status older AML patients with DNMT3A mutations and showed that MRD-negativity did not predict OS and LFS in DNMT3A-mut patients. The manuscript is well-written and of clinical relevance. I only have the following minor comments:

- Methods: A paragraph on how MRD status was assessed would be helpful. I see that the authors refer to a prior publication (line 108) but it would be easier for the reader if there was a brief 1-2 sentence description in the manuscript at hand as well.

- Results; baseline patient characteristics: Do the authors have information available on how many patients with DNMT3A mutations had sAML and in particular AML-MRC and how AML-MRC status compared with de novo AML in terms of outcome and prognostic impact of MRD status.

- Results: Do the authors have any data on a potential impact of DNMT3A variant allele frequency? Was the DNMT3A mutation part of the predominant clone or rather a CHIP clone?

Author Response

Thank you for your comments. 

- Methods: A paragraph on how MRD status was assessed would be helpful. I see that the authors refer to a prior publication (line 108) but it would be easier for the reader if there was a brief 1-2 sentence description in the manuscript at hand as well.

-->Manuscript have been improved with a quick description and references regarding MRD assessment (line 117-121).

- Results; baseline patient characteristics: Do the authors have information available on how many patients with DNMT3A mutations had sAML and in particular AML-MRC and how AML-MRC status compared with de novo AML in terms of outcome and prognostic impact of MRD status.

-->The number of patients with documented history of sAML have been implemented in the main text (line 149) and in table 1. Due to the retrospective nature of the study, we were not able to rule out all MRC criterias in our cytology archives.

- Results: Do the authors have any data on a potential impact of DNMT3A variant allele frequency? Was the DNMT3A mutation part of the predominant clone or rather a CHIP clone?

--> DNMT3A mutations were usually part of the main clone with 42% median VAF. Only two patients exert VAF < 10% that might be related to a CHIP. Table 1 have been implemented with VAF and shortly described in the main text (line 152)

General comment

Introduction have been improved (line 34-48) and a simple summary (line 24-33) added to the manuscript.

Round 2

Reviewer 2 Report

Changes performed by the authors have improved the article. Minor revisions required.